# LEARNING TO SCHEDULE COMMUNICATION IN MULTI-AGENT REINFORCEMENT LEARNING

**Daewoo Kim, Sangwoo Moon, David Hostallero, Wan Ju Kang, Taeyoung Lee,**
**Kyunghwan Son & Yung Yi**
School of Electrical Engineering, KAIST
Daejeon, South Korea
`{dwkim, swmoon, ddhostallero, wjkang, taeyoung.lee, khson}@lanada.kaist.`
`ac.kr, yiyung@kaist.edu`

## ABSTRACT

Many real-world reinforcement learning tasks require multiple agents to make sequential decisions under the agents' interaction, where well-coordinated actions among the agents are crucial to achieve the target goal better at these tasks. One way to accelerate the coordination effect is to enable multiple agents to communicate with each other in a distributed manner and behave as a group. In this paper, we study a practical scenario when *(i)* the communication bandwidth is limited and *(ii)* the agents share the communication medium so that only a restricted number of agents are able to simultaneously use the medium, as in the state-of-the-art wireless networking standards. This calls for a certain form of *communication scheduling*. In that regard, we propose a multi-agent deep reinforcement learning framework, called SchedNet, in which agents learn how to schedule themselves, how to encode the messages, and how to select actions based on received messages. SchedNet is capable of deciding which agents should be entitled to broadcasting their (encoded) messages, by learning the importance of each agent's partially observed information. We evaluate SchedNet against multiple baselines under two different applications, namely, cooperative communication and navigation, and predator-prey. Our experiments show a non-negligible performance gap between SchedNet and other mechanisms such as the ones without communication and with vanilla scheduling methods, *e.g.*, round robin, ranging from 32% to 43%.

## 1 INTRODUCTION

Reinforcement Learning (RL) has garnered renewed interest in recent years. Playing the game of Go (Mnih et al., 2015), robotics control (Gu et al., 2017; Lillicrap et al., 2015), and adaptive video streaming (Mao et al., 2017) constitute just a few of the vast range of RL applications. Combined with developments in deep learning, deep reinforcement learning (Deep RL) has emerged as an accelerator in related fields. From the well-known success in single-agent deep reinforcement learning, such as Mnih et al. (2015), we now witness growing interest in its multi-agent extension, the multi-agent reinforcement learning (MARL), exemplified in Gupta et al. (2017); Lowe et al. (2017); Foerster et al. (2017a); Omidshafiei et al. (2017); Foerster et al. (2016); Sukhbaatar et al. (2016); Mordatch & Abbeel (2017); Havrylov & Titov (2017); Palmer et al. (2017); Peng et al. (2017); Foerster et al. (2017c); Tampuu et al. (2017); Leibo et al. (2017); Foerster et al. (2017b). In the MARL problem commonly addressed in these works, multiple agents interact in a single environment repeatedly and improve their policy iteratively by learning from observations to achieve a common goal. Of particular interest is the distinction between two lines of research: one fostering the direct communication among agents themselves, as in Foerster et al. (2016); Sukhbaatar et al. (2016) and the other coordinating their cooperative behavior without direct communication, as in Foerster et al. (2017b); Palmer et al. (2017); Leibo et al. (2017).

In this work, we concern ourselves with the former. We consider MARL scenarios wherein the task at hand is of a cooperative nature and agents are situated in a partially observable environment, but each endowed with different observation power. We formulate this scenario into a multi-agent

sequential decision-making problem, such that all agents share the goal of maximizing the same discounted sum of rewards. For the agents to directly communicate with each other and behave as a coordinated group rather than merely coexisting individuals, they must carefully determine the information they exchange under a practical bandwidth-limited environment and/or in the case of high-communication cost. To coordinate this exchange of messages, we adopt the centralized training and distributed execution paradigm popularized in recent works, *e.g.*, Foerster et al. (2017a); Lowe et al. (2017); Sunehag et al. (2018); Rashid et al. (2018); Gupta et al. (2017).

In addition to bandwidth-related constraints, we take the issues of sharing the communication medium into consideration, especially when agents communicate over wireless channels. The state-of-the-art standards on wireless communication such as Wi-Fi and LTE specify the way of scheduling users as one of the basic functions. However, as elaborated in Related work, MARL problems involving scheduling of only a restricted set of agents have not yet been extensively studied. The key challenges in this problem are: *(i)* that limited bandwidth implies that agents must exchange succinct information: something concise and yet meaningful and *(ii)* that the shared medium means that potential contenders must be appropriately arbitrated for proper collision avoidance, necessitating a certain form of communication scheduling, popularly referred to as MAC (Medium Access Control) in the area of wireless communication. While stressing the coupled nature of the encoding/decoding and the scheduling issue, we zero in on the said communication channel-based concerns and construct our neural network accordingly.

**Contributions** In this paper, we propose a new deep multi-agent reinforcement learning architecture, called SchedNet, with the rationale of centralized training and distributed execution in order to achieve a common goal better via decentralized cooperation. During distributed execution, agents are allowed to communicate over wireless channels where messages are broadcast to all agents in each agent's communication range. This broadcasting feature of wireless communication necessitates a Medium Access Control (MAC) protocol to arbitrate contending communicators in a shared medium. CSMA (Collision Sense Multiple Access) in Wi-Fi is one such MAC protocol. While prior work on MARL to date considers only the limited bandwidth constraint, we additionally address the shared medium contention issue in what we believe is the first work of its kind: which nodes are granted access to the shared medium. Intuitively, nodes with more important observations should be chosen, for which we adopt a simple yet powerful mechanism called weight-based scheduler (WSA), designed to reconcile simplicity in training with integrity of reflecting real-world MAC protocols in use (*e.g.*, 802.11 Wi-Fi). We evaluate SchedNet for two applications: cooperative communication and navigation and predator/prey and demonstrate that SchedNet outperforms other baseline mechanisms such as the one without any communication or with a simple scheduling mechanism such as round robin. We comment that SchedNet is not intended for competing with other algorithms for cooperative multi-agent tasks without considering scheduling, but a complementary one. We believe that adding our idea of agent scheduling makes those algorithms much more practical and valuable.

**Related work** We now discuss the body of relevant literature. Busoniu et al. (2008) and Tan (1993) have studied MARL with decentralized execution extensively. However, these are based on tabular methods so that they are restricted to simple environments. Combined with developments in deep learning, deep MARL algorithms have emerged (Tampuu et al., 2017; Foerster et al., 2017a; Lowe et al., 2017). Tampuu et al. (2017) uses a combination of DQN with independent Q-learning. This independent learning does not perform well because each agent considers the others as a part of environment and ignores them. Foerster et al. (2017a); Lowe et al. (2017); Gupta et al. (2017); Sunehag et al. (2018), and Foerster et al. (2017b) adopt the framework of centralized training with decentralized execution, empowering the agent to learn cooperative behavior considering other agents' policies without any communication in distributed execution.

It is widely accepted that communication can further enhance the collective intelligence of learning agents in their attempt to complete cooperative tasks. To this end, a number of papers have previously studied the learning of communication protocols and languages to use among multiple agents in reinforcement learning. We explore those bearing the closest resemblance to our research. Foerster et al. (2016); Sukhbaatar et al. (2016); Peng et al. (2017); Guestrin et al. (2002), and Zhang & Lesser (2013) train multiple agents to learn a communication protocol, and have shown that communicating agents achieve better rewards at various tasks. Mordatch & Abbeel (2017) and Havrylov & Titov (2017) investigate the possibility of the artificial emergence of language. Coordinated RL by Guestrin et al. (2002) is an earlier work demonstrating the feasibility of structured communication and the agents' selection of jointly optimal action.

Only DIAL (Foerster et al., 2016) and Zhang & Lesser (2013) explicitly address bandwidth-related concerns. In DIAL, the communication channel of the training environment has a limited bandwidth, such that the agents being trained are urged to establish more resource-efficient communication protocols. The environment in Zhang & Lesser (2013) also has a limited-bandwidth channel in effect, due to the large amount of exchanged information in running a distributed constraint optimization algorithm. Recently, Jiang & Lu (2018) proposes an attentional communication model that allows some agents who request additional information from others to gather observation from neighboring agents. However, they do not explicitly consider the constraints imposed by limited communication bandwidth and/or scheduling due to communication over a shared medium.

To the best of our knowledge, there is no prior work that incorporates an intelligent scheduling entity in order to facilitate inter-agent communication in both a limited-bandwidth and shared medium access scenarios. As outlined in the introduction, intelligent scheduling among learning agents is pivotal in the orchestration of their communication to better utilize the limited available bandwidth as well as in the arbitration of agents contending for shared medium access.

## 2 BACKGROUND

**Reinforcement Learning** We consider a standard RL formulation based on Markov Decision Process (MDP). An MDP is a tuple $< \mathcal{S}, \mathcal{A}, r, P, \gamma >$ where $\mathcal{S}$ and $\mathcal{A}$ are the sets of states and actions, respectively, and $\gamma \in [0, 1]$ is the discount factor. A transition probability function $P : \mathcal{S} \times \mathcal{A} \rightarrow \mathcal{S}$ maps states and actions to a probability distribution over next states, and $r : \mathcal{S} \times \mathcal{A} \rightarrow \mathbb{R}$ denotes the reward. The goal of RL is to learn a policy $\pi : \mathcal{S} \rightarrow \mathcal{A}$ that solves the MDP by maximizing the expected discounted return $R_t = \mathbb{E}[\sum_{k=0}^{\infty} \gamma^k r_{t+k}|\pi]$. The policy induces a value function $V^\pi(s) = \mathbb{E}_\pi[R_t|s_t = s]$, and an action value function $Q^\pi(s, a) = \mathbb{E}_\pi[R_t|s_t = s, a_t = a]$.

**Actor-critic Method** The main idea of the policy gradient method is to optimize the policy, parametrized by $\theta^\pi$, in order to maximize the objective $J(\theta) = \mathbb{E}_{s \sim p^\pi, a \sim \pi_\theta}[R]$ by directly adjusting the parameters in the direction of the gradient. By the policy gradient theorem Sutton et al. (2000), the gradient of the objective is:

$$\nabla_\theta J(\pi_\theta) = \mathbb{E}_{s \sim \rho^\pi, a \sim \pi_\theta}[\nabla_\theta \log \pi_\theta(a|s)Q^\pi(s, a)], \quad (1)$$

where $\rho^\pi$ is the state distribution. Our baseline algorithmic framework is the *actor-critic* approach Konda & Tsitsiklis (2003). In this approach, an *actor* adjusts the parameters $\theta$ of the policy $\pi_\theta(s)$ by gradient ascent. Instead of the unknown true action-value function $Q^\pi(s, a)$, its approximated version $Q^w(s, a)$ is used with parameter $w$. A *critic* estimates the action-value function $Q^w(s, a)$ using an appropriate policy evaluation algorithm such as temporal-difference learning Tesauro (1995). To reduce the variance of the gradient updates, some baseline function $b(s)$ is often subtracted from the action value, thereby resulting in $Q^\pi(s, a) - b(s)$ Sutton & Barto (1998). A popular choice for this baseline function is the state value $V(s)$, which indicates the inherent "goodness" of the state. This difference between the action value and the state value is often dubbed as the advantage $A(s, a)$ whose TD-error-based substitute $\delta_t = r_t + \gamma V(s_{t+1}) - V(s_t)$ is an unbiased estimate of the advantage as in Mnih et al. (2016). The actor-critic algorithm can also be applied to a deterministic policy $\mu_\theta : \mathcal{S} \rightarrow \mathcal{A}$. By the deterministic policy gradient theorem Silver et al. (2014), we update the parameters as follows:

$$\nabla_\theta J(\mu_\theta) = \mathbb{E}_{s \sim \rho^\mu}[\nabla_\theta \mu_\theta(s)\nabla_a Q^\mu(s, a)|_{a=\mu_\theta(s)}]. \quad (2)$$

**MARL: Centralized Critic and Distributed Actor (CCDA)** We formalize MARL using DEC-POMDP (Oliehoek et al., 2016), which is a generalization of MDP to allow a distributed control by multiple agents who may be incapable of observing the global state. A DEC-POMDP is described by a tuple $< \mathcal{S}, \mathcal{A}, r, P, \Omega, \mathcal{O}, \gamma >$. We use bold-face fonts in some notations to highlight the context of multi-agents. Each agent $i \in \mathcal{N}$ chooses an action $a_i \in \mathcal{A}$, forming a joint action vector $\boldsymbol{a} = [a_i] \in \mathcal{A}^n$ and has partial observations $o_i \in \Omega$ according to some observation function $\mathcal{O}(\boldsymbol{s}, i) : \mathcal{S} \times \mathcal{N} \mapsto \Omega$. $P(\boldsymbol{s'}|\boldsymbol{s}, \boldsymbol{a}) : \mathcal{S} \times \mathcal{A}^n \mapsto [0, 1]$ is the transition probability function. All agents share the same reward $r(\boldsymbol{s}, \boldsymbol{u}) : \mathcal{S} \times \mathcal{A}^n \mapsto \mathbb{R}$. Each agent $i$ takes action $a_i$ based on its own policy $\pi^i(a_i|o_i)$. As mentioned in Section 1, our particular focus is on the centralized training and distributed execution paradigm, where the actor-critic approach is a good fit to such a paradigm. Since the agents should execute in a distributed setting, each agent, say $i$, maintains its own actor

that selects $i$'s action based only on what is partially observed by $i$. The critic is naturally responsible for centralized training, and thus works in a centralized manner. Thus, the critic is allowed to have the global state $s$ as its input, which includes all agents' observations and extra information from the environment. The role of the critic is to "criticize" individual agent's actions. This centralized nature of the critic helps in providing more accurate feedback to the individual actors with limited observation horizon. In this case, each agent's policy, $\pi^i$, is updated by a variant of (1) as:

$$\nabla_\theta J(\pi_\theta^i) = \mathbb{E}_{s \sim \rho^\pi, a \sim \pi_\theta}[\nabla_\theta \log \pi_\theta^i(a_i|o_i)(r + \gamma V(s_{t+1}) - V(s_t))]. \tag{3}$$

## 3 METHOD

### 3.1 COMMUNICATION ENVIRONMENT AND PROBLEM

In practical scenarios where agents are typically separated but are able to communicate over a shared medium, *e.g.*, a frequency channel in wireless communications, two important constraints are imposed: bandwidth and contention for medium access (Rappaport, 2001). The bandwidth constraint entails a limited amount of bits per unit time, and the contention constraint involves having to avoid collision among multiple transmissions due to the natural aspect of signal broadcasting in wireless communication. Thus, only a restricted number of agents are allowed to transmit their messages each time step for a reliable message transfer. In this paper, we use a simple model to incorporate that the aggregate information size per time step is limited by $L_{\text{band}}$ bits and that only $K_{\text{sched}}$ out of $n$ agents may broadcast their messages.

**Weight-based Scheduling** Noting that distributed execution of agents is of significant importance, there may exist a variety of scheduling mechanisms to schedule $K_{\text{sched}}$ agents in a distributed manner. In this paper, we adopt a simple algorithm that is weight-based, which we call WSA (Weight-based Scheduling Algorithm). Once each agent decides its own weight, the agents are scheduled based on their weights following a class of the pre-defined rules. We consider the following two specific ones among many different proposals due to simplicity, but more importantly, good approximation of wireless scheduling protocols in practice.

○ *Top(k)*. Selecting top $k$ agents in terms of their weight values.
○ *Softmax(k)*. Computing softmax values $\sigma_i(\boldsymbol{w}) = \frac{e^{w_i}}{\sum_{j=1}^n e^{w_j}}$ for each agent $i$, and then randomly selecting $k$ agents acoording to the probability distribution $[\sigma_i(\boldsymbol{w})]_{i=1}^n$.

Since distributed execution is one of our major operational constraints in SchedNet or other CTDE-based MARL algorithms, Top($k$) and Softmax($k$) should be realizable via a weight-based mechanism in a distributed manner. In fact, this has been an active research topic to date in wireless networking, where many algorithms exist (Tassiulas & Ephremides, 1992; Yi et al., 2008; Jiang & Walrand, 2010). Due to space limitation, we present how to obtain distributed versions of those two rules based on weights in our supplementary material. To summarize, using so-called CSMA (Carrier Sense Multiple Access) (Kurose, 2005), which is a fully distributed MAC scheduler and forms a basis of Wi-Fi, given agents' weight values, it is possible to implement Top($k$) and Softmax($k$).

Our goal is to train agents so that every time each agent takes an action, only $K_{\text{sched}}$ agents can broadcast their messages with limited size $L_{\text{band}}$ with the goal of receiving the highest cumulative reward via cooperation. Each agent should determine a policy described by its scheduling weights, encoded communication messages, and actions.

### 3.2 ARCHITECTURE

To this end, we propose a new deep MARL framework with scheduled communications, called SchedNet, whose overall architecture is depicted in Figure 1. SchedNet consists of the following three components: *(i)* actor network, *(ii)* scheduler, and *(iii)* critic network. This section is devoted to presenting the architecture only, whose details are presented in the subsequent sections.

**Neural networks** The actor network is the collection of $n$ per-agent individual actor networks, where each agent $i$'s individual actor network consists of a triple of the following networks: a

message encoder, an action selector, and a weight generator, as specified by:

message encoder $f_{\mathrm{enc}}^i : o_i \mapsto m_i,$

action selector $f_{\mathrm{as}}^i : (o_i, \boldsymbol{m} \otimes \boldsymbol{c}) \mapsto u_i,$

weight generator $f_{\mathrm{wg}}^i : o_i \mapsto w_i.$

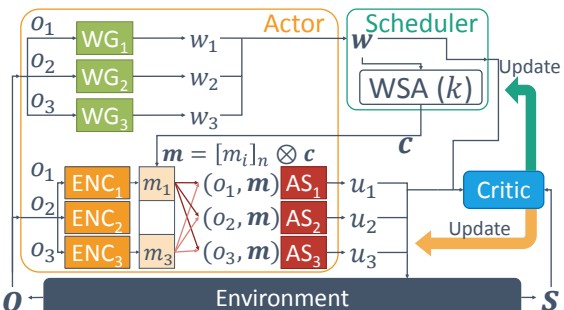

Here, $\boldsymbol{m} = [m_i]_n{}^1$ is the vector of each $i$'s encoded message $m_i$. An agent schedule vector $\boldsymbol{c} = [c_i]_n, c_i \in \{0,1\}$ represents whether each agent is scheduled. Note that agent $i$'s encoded message $m_i$ is generated by a neural network $f_{\mathrm{enc}}^i : o_i \mapsto m_i$. The operator "$\otimes$" concatenates all the sched-

Figure 1: Architecture of SchedNet with three agents. Agents 1 and 3 have been scheduled for this time step.

uled agents' messages. For example, for $\boldsymbol{m} = [010, 111, 101]$ and $\boldsymbol{c} = [110]$, $\boldsymbol{m} \otimes \boldsymbol{c} = 010111$. This concatenation with the schedule profile $\boldsymbol{c}$ means that only those agents scheduled in $\boldsymbol{c}$ may broadcast their messages to all other agents. We denote by $\theta_{\mathrm{as}}^i$, $\theta_{\mathrm{wg}}^i$, and $\theta_{\mathrm{enc}}^i$ the parameters of the action selector, the weight generator, and the encoder of agent $i$, respectively, where we let $\boldsymbol{\theta}_{\mathrm{as}} = [\theta_{\mathrm{as}}^i]_n$, and similarly define $\boldsymbol{\theta}_{\mathrm{wg}}$ and $\boldsymbol{\theta}_{\mathrm{enc}}$.

**Coupling: Actor and Scheduler** Encoder, weight generator and the scheduler are the modules for handling the constraints of limited bandwidth and shared medium access. Their common goal is to learn the state-dependent "importance" of individual agent's observation, encoders for generating compressed messages and the scheduler for being used as a basis of an external scheduling mechanism based on the weights generated by per-agent weight generators. These three modules work together to smartly respond to time-varying states. The action selector is trained to decode the incoming message, and consequently, to take a good action for maximizing the reward. At every time step, the schedule profile $\boldsymbol{c}$ varies depending on the observation of each agent, so the incoming message $\boldsymbol{m}$ comes from a different combination of agents. Since the agents can be heterogeneous and they have their own encoder, the action selector must be able to make sense of incoming messages from different senders. However, the weight generator's policy changes, the distribution of incoming messages also changes, which is in turn affected by the pre-defined WSA. Thus, the action selector should adjust to this changed scheduling. This also affects the encoder in turn. The updates of the encoder and the action selector trigger the update of the scheduler again. Hence, weight generators, message encoders, and action selectors are strongly coupled with dependence on a specific WSA, and we train those three networks at the same time with a common critic.

**Scheduling logic** The schedule profile $\boldsymbol{c}$ is determined by the WSA module, which is mathematically a mapping from all agents' weights $\boldsymbol{w}$ (generated by $f_{\mathrm{wg}}^i$) to $\boldsymbol{c}$. Typical examples of these mappings are *Top(k)* and *Softmax(k)*, as mentioned above. The scheduler of each agent is trained appropriately depending on the employed WSA algorithm.

## 3.3 TRAINING AND EXECUTION

In the centralized training with distributed execution, for a given WSA, we include all components and modules in Figure 1 to search for $\boldsymbol{\theta}_{\mathrm{as}}$, $\boldsymbol{\theta}_{\mathrm{wg}}$, and $\boldsymbol{\theta}_{\mathrm{enc}}$, whereas in execution, each agent $i$ runs a certain shared medium access mechanism, well-modeled by a weight-based scheduler, and just needs three agent-specific parameters $\theta_{\mathrm{as}}^i$, $\theta_{\mathrm{wg}}^i$, and $\theta_{\mathrm{enc}}^i$.

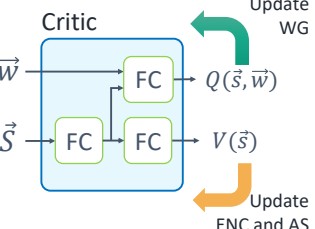

### 3.3.1 CENTRALIZED TRAINING

**Centralized critic** The actor is trained by dividing it into two parts: *(i)* message encoders and action selectors, and *(ii)* weight generators. This partitioning is motivated by the fact that it is hard to update both parts with one backpropagation since WSA is not differentiable. To update the actor, we use a centralized critic

Figure 2: Architecture of the critic. FC stands for fully connected neural network.

---

[1]We use $[\cdot]_n$ to mean the $n$-dimensional vector, where $n$ is the number of agents.

parametrized by $\theta_c$ to estimate the state value function $V_{\theta_c}(\boldsymbol{s})$ for the action selectors and message encoders, and the action-value function $Q_{\theta_c}^\pi(\boldsymbol{s}, \boldsymbol{w})$ for the weight generators. The critic is used only when training, and it can use the global state $\boldsymbol{s}$, which includes the observation of all agents. All networks in the actor are trained with gradient-based on temporal difference backups. To share common features between $V_{\theta_c}(\boldsymbol{s})$ and $Q_{\theta_c}^\pi(\boldsymbol{s}, \boldsymbol{w})$ and perform efficient training, we use shared parameters in the lower layers of the neural network between the two functions, as shown in Figure 2.

**Weight generators**   We consider the collection of all agents' WGs as a single neural network $\mu_{\boldsymbol{\theta}_{\mathrm{wg}}}(\boldsymbol{o})$ mapping from $\boldsymbol{o}$ to $\boldsymbol{w}$, parametrized by $\boldsymbol{\theta}_{\mathrm{wg}}$. Noting that $w_i$ is a continuous value, we apply the DDPG algorithm (Lillicrap et al., 2015), where the entire policy gradient of the collection of WGs is given by:

$$\nabla_{\boldsymbol{\theta}_{\mathrm{wg}}} J(\boldsymbol{\theta}_{\mathrm{wg}}, \cdot) = \mathbb{E}_{\boldsymbol{w} \sim \mu_{\boldsymbol{\theta}_{\mathrm{wg}}}}[\nabla_{\boldsymbol{\theta}_{\mathrm{wg}}} \mu_{\boldsymbol{\theta}_{\mathrm{wg}}}(\boldsymbol{o}) \nabla_{\boldsymbol{w}} Q_{\theta_c}(\boldsymbol{s}, \boldsymbol{w})|_{\boldsymbol{w} = \mu_{\boldsymbol{\theta}_{\mathrm{wg}}}(\boldsymbol{o})}].$$

We sample the policy gradient for sufficient amount of experience in the set of all scheduling profiles, i.e., $\mathcal{C} = \{\boldsymbol{c} \mid \sum_{c_i} \leq k\}$. The values of $Q_{\theta_c}(\boldsymbol{s}, \boldsymbol{w})$ are estimated by the centralized critic, where $\boldsymbol{s}$ is the global state corresponding to $\boldsymbol{o}$ in a sample.

**Message encoders and action selectors**   The observation of each agent travels through the encoder and the action selector. We thus serialize $f_{\mathrm{enc}}^i$ and $f_{\mathrm{as}}^i$ together and merge the encoders and actions selectors of all agents into one aggregate network $\pi_{\boldsymbol{\theta}_u}(\boldsymbol{u}|\boldsymbol{o}, \boldsymbol{c})$, which is parametrized by $\boldsymbol{\theta}_u = \{\boldsymbol{\theta}_{\mathrm{enc}}, \boldsymbol{\theta}_{\mathrm{as}}\}$. This aggregate network $\pi_{\boldsymbol{\theta}_u}$ learns via backpropagation of actor-critic policy gradients, described below. The gradient of this objective function, which is a variant of (3), is given by

$$\nabla_{\boldsymbol{\theta}_u} J(\cdot, \boldsymbol{\theta}_u) = \mathbb{E}_{\boldsymbol{s} \sim \rho^\pi, \boldsymbol{u} \sim \pi_{\boldsymbol{\theta}_u}}[\nabla_{\boldsymbol{\theta}_u} \log \pi_{\boldsymbol{\theta}_u}(\boldsymbol{u}|\boldsymbol{o}, \boldsymbol{c})[r + \gamma V_{\theta_c}(\boldsymbol{s}') - V_{\theta_c}(\boldsymbol{s})]], \tag{4}$$

where $\boldsymbol{s}$ and $\boldsymbol{s}'$ are the global states corresponding to the observations at current and next time step. We can get the value of state $V_{\theta_c}(\boldsymbol{s})$ from the centralized critic and then adjust the parameters $\boldsymbol{\theta}_u$ via gradient ascent accordingly.

### 3.3.2   DISTRIBUTED EXECUTION

In execution, each agent $i$ should be able to determine the scheduling weight $w_i$, encoded message $m_i$, and action selection $u_i$ in a distributed manner. This process must be based on its own observation, and the weights generated by its own action selector, message encoder, and weight generator with the parameters $\theta_{\mathrm{as}}^i$, $\theta_{\mathrm{enc}}^i$, and $\theta_{\mathrm{wg}}^i$, respectively. After each agent determines its scheduling weight, $K_{\mathrm{sched}}$ agents are scheduled by WSA, which leads the encoded messages of scheduled agents to be broadcast to all agents. Finally, each agent finally selects an action by using received messages. This process is sequentially repeated under different observations over time.

## 4   EXPERIMENT

**Environments**   To evaluate SchedNet[2], we consider two different environments for demonstrative purposes: Predator and Prey (PP) which is used in Stone & Veloso (2000), and Cooperative Communication and Navigation (CCN) which is the simplified version of the one in Lowe et al. (2017). The detailed experimental environments are elaborated in the following subsections as well as in supplementary material. We take the communication environment into our consideration as follows. $k$ out of all agents can have the chance to broadcast the message whose bandwidth[3] is limited by $l$.

**Tested algorithms and setup**   We perform experiments in aforementioned environments. We compare SchedNet with a variant of DIAL,[4] (Foerster et al., 2016) which allows communication with limited bandwidth. During the execution of DIAL, the limited number ($k$) of agents are scheduled following a simple round robin scheduling algorithm, and the agent reuses the outdated messages of non-scheduled agents to make a decision on the action to take, which is called DIAL($k$). The other baselines are independent DQN (IDQN) (Tampuu et al., 2017) and COMA (Foerster et al., 2017a) in which no agent is allowed to communicate. To see the impact of scheduling in SchedNet, we

---

[2]The code is available on `https://github.com/rhoowd/sched_net`
[3]The unit of bandwidth is 2 bytes which can express one real value (float16 type)
[4]We train and execute DIAL without discretize/regularize unit (DRU), because in our setting, agents can exchange messages that can express real values.

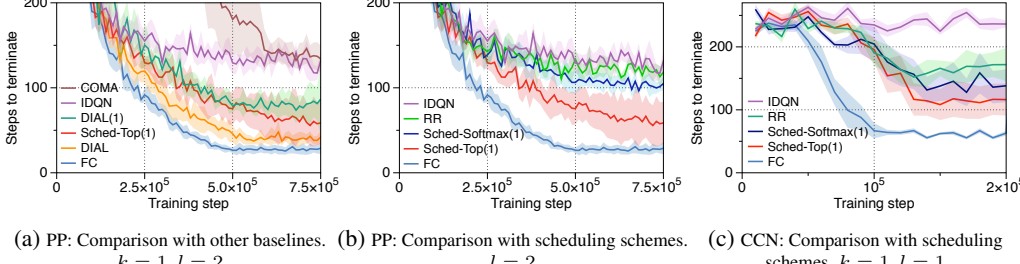

(a) PP: Comparison with other baselines. $k = 1, l = 2$

(b) PP: Comparison with scheduling schemes. $l = 2$

(c) CCN: Comparison with scheduling schemes. $k = 1, l = 1$

Figure 3: Learning curves during the learning of the PP and CCN tasks. The plots show the average time taken to complete the task, where shorter time is better for the agents.

compare SchedNet with *(i)* RR (round robin), which is a canonical scheduling method in communication systems where all agents are sequentially scheduled, and *(ii)* FC (full communication), which is the ideal configuration, wherein all the agents can send their messages without any scheduling or bandwidth constraints. We also diversify the WSA in SchedNet into: *(i)* Sched-Softmax(1) and *(ii)* Sched-Top(1) whose details are in Section 3.1. We train our models until convergence, and then evaluate them by averaging metrics for 1,000 iterations. The shaded area in each plot denotes 95% confidence intervals based on 6-10 runs with different seeds.

## 4.1 PREDATOR AND PREY

In this task, there are multiple agents who must capture a randomly moving prey. Agents' observations include position of themselves and the relative positions of prey, if observed. We employ four agents, and they have different observation horizons, where only agent 1 has a $5 \times 5$ view while agents 2, 3, and 4 have a smaller, $3 \times 3$ view. The predators are rewarded when they capture the prey, and thus the performance metric is the number of time steps taken to capture the prey.

**Result in PP** Figure 3a illustrates the learning curve of 750,000 steps in PP. In FC, since the agents can use full state information even during execution, they achieve the best performance. SchedNet outperforms IDQN and COMA in which communication is not allowed. It is observed that agents first find the prey, and then follow it until all other agents also eventually observe the prey. An agent successfully learns to follow the prey after it observes the prey but that it takes a long time to meet the prey for the first time. If the agent broadcasts a message that includes the location information of the prey, then other agents can find the prey more quickly. Thus, it is natural that SchedNet and DIAL perform better than IDQN or COMA, because they are trained to work with communication. However, DIAL is not trained for working under medium contention constraints. Although DIAL works well when there is no contention constraints, under the condition where only one agent is scheduled to broadcast the message by a simple scheduling algorithm (*i.e.*, RR), the average number of steps to capture the prey in DIAL(1) is larger than that of SchedNet-Top(1), because the outdated messages of non-scheduled agents is noisy for the agents to decide on actions. Thus, we should consider the scheduling from when we train the agents to make them work in a demanding environment.

**Impact of intelligent scheduling** In Figure 3b, we observe that IDQN, RR, and SchedNet-Softmax(1) lie more or less on a comparable performance tier, with SchedNet-Softmax(1) as the best in the tier. SchedNet-Top(1) demonstrates a non-negligible gap better than the said tier, implying that a deterministic selection improves the agents' collective rewards the best. In particular, SchedNet-Top(1) improves the performance by 43% compared to RR. Figure 3b lets us infer that, while all the agents are trained under the same conditions except for the scheduler, the difference in the scheduler is the sole determining factor for the variation in the performance levels. Thus, ablating away the benefit from smart encoding, the intelligent scheduling element in SchedNet can be accredited with the better performance.

**Weight-based Scheduling** We attempt to explain the internal behavior of SchedNet by investigating instances of temporal scheduling profiles obtained during the execution. We observe that SchedNet has learned to schedule those agents with a farther observation horizon, realizing the rationale of importance-based assignment of scheduling priority also for the PP scenario. Recall that Agent 1 has a wider view and thus tends to obtain valuable observation more frequently. In Fig-

ure 4, we see that scheduling chances are distributed over (14, 3, 4, 4) where corresponding average weights are (0.74, 0.27, 0.26, 0.26), implying that those with greater observation power tend to be scheduled more often.

**Message encoding** We now attempt to understand what the predator agents communicate when performing the task. Figure 5 shows the projections of the messages onto a 2D plane, which is generated by the scheduled agent under SchedNet-Top(1) with $l = 2$. When the agent does not observe the prey (blue circle in Figure), most of the messages reside in the bottom or the left partition of the plot. On the other hand, the messages have large variance when it observes the prey (red 'x'). This is because the agent should transfer more informative messages that implicitly include the location of the prey, when it observes the prey. Further analysis of the messages is presented in our supplementary material.

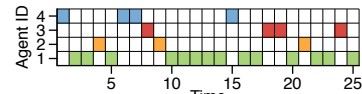

Figure 4: Instances of scheduling results over 25 time steps in PP

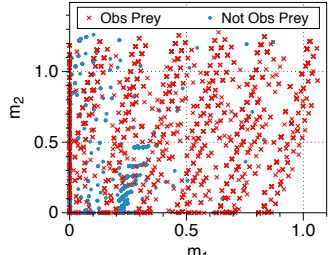

Figure 5: Encoded messages projected onto 2D plane in PP task

## 4.2 COOPERATIVE COMMUNICATION AND NAVIGATION

In this task, each agent's goal is to arrive at a pre-specified destination on its one-dimensional world, and they collect a joint reward when both agents reach their respective destination. Each agent has a zero observation horizon around itself, but it can observe the situation of the other agent. We introduce heterogeneity into the scenario, where the agent-destination distance at the beginning of the task differs across agents. The metric used to gauge the performance is the number of time steps taken to complete the CCN task.

**Result in CCN** We examine the CCN environment whose results are shown in Figure 3c. Sched-Net and other baselines were trained for 200,000 steps. As expected, IDQN takes the longest time, and FC takes the shortest time. RR exhibits mediocre performance, better than IDQN, because agents at least take turns in obtaining the communication opportunity. Of particular interest is Sched-Net, outperforming both IDQN and RR with a non-negligible gap. We remark that the deterministic selection with SchedNet-Top(1) slightly beats the probabilistic counterpart, SchedNet-Softmax(1). The 32% improved gap between RR and SchedNet clearly portrays the effects of intelligent scheduling, as the carefully learned scheduling method of SchedNet was shown to complete the CCN task faster than the simplistic RR.

**Scheduling in CCN** As Agent 2 is farther from its destination than Agent 1, we observe that Agent 1 is scheduled more frequently to drive Agent 2 to its destination (7 vs. 18), as shown in Figure 6. This evidences that SchedNet flexibly adapts to heterogeneity of agents via scheduling. Towards more efficient completion of the task, a rationale of *more scheduling for more important agents* should be implemented. This is in accordance with the results obtained from PP environments: more important agents are scheduled more.

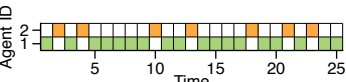

Figure 6: Instances of scheduling results over 25 time steps in CCN

## 5 CONCLUSION

We have proposed SchedNet for learning to schedule inter-agent communications in fully-cooperative multi-agent tasks. In SchedNet, we have the centralized critic giving feedback to the actor, which consists of message encoders, action selectors, and weight generators of each individual agent. The message encoders and action selectors are criticized towards compressing observations more efficiently and selecting actions that are more rewarding in view of the cooperative task at hand. Meanwhile, the weight generators are criticized such that $k$ agents with apparently more valuable observation are allowed to access the shared medium and broadcast their messages to all other agents. Empirical results and an accompanying ablation study indicate that the learnt encoding and scheduling behavior each significantly improve the agents' performance. We have observed that an intelligent, distributed communication scheduling can aid in a more efficient, coordinated, and rewarding behavior of learning agents in the MARL setting.

ACKNOWLEDGE

This work was supported by Institute for Information communications Technology Promotion(IITP) grant funded by the Korea government(MSIT) (No.2018-0-00170, Virtual Presence in Moving Objects through 5G)

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

## SUPPLEMENTARY MATERIAL

## A   SCHEDNET TRAINING ALGORITHM

The training algorithm for SchedNet is provided in Algorithm 1. The parameters of the message encoder are assumed to be included in the actor network. Thus, we use the notation $f_{\mathrm{as}}^i(o^i, \boldsymbol{c}) = f_{\mathrm{as}}^i(o^i, f_{\mathrm{enc}}^i(o^i) \otimes \boldsymbol{c})$ to simplify the presentation.

---

**Algorithm 1** SchedNet

1: Initialize actor parameters $\theta_u$, scheduler parameters $\theta_{\mathrm{wg}}$, and critic parameters $\theta_c$
2: Initialize target scheduler parameters $\theta'_{\mathrm{wg}}$, and target critic parameters $\theta'_c$
3: **for** episode = 1 to $M$ **do**
4:     Observe initial state $\boldsymbol{s}$
5:     **for** $t = 1$ to $T$ **do**
6:         $\boldsymbol{w}_t \leftarrow$ the priority $w^i = f_{\mathrm{wg}}^i(o^i)$ of each agent $i$
7:         Get schedule $\boldsymbol{c_t}$ from $\boldsymbol{w}_t$
8:         $\boldsymbol{u}_t \leftarrow$ the action $u^i = f_{\mathrm{as}}^i(o^i, \boldsymbol{c_t})$ of each agent $i$
9:         Execute the actions $\boldsymbol{u}_t$ and observe the reward $r_t$ and next state $\boldsymbol{s}_{t+1}$
10:        Store $(\boldsymbol{s}_t, \boldsymbol{u}_t, r_t, \boldsymbol{s}_{t+1}, \boldsymbol{c_t}, \boldsymbol{w}_t)$ in the replay buffer $B$
11:        Sample a minibatch of $S$ samples $(\boldsymbol{s}_k, \boldsymbol{u}_k, r_k, \boldsymbol{s}_{k+1}, \boldsymbol{c_k}, \boldsymbol{w}_k)$ from $B$
12:        Set $y_k = r_k + \gamma \bar{V}(\boldsymbol{s}_{k+1})$
13:        Set $\hat{y}_k = r_k + \gamma \bar{Q}(\boldsymbol{s}_{k+1}, \bar{f}_{\mathrm{wg}}^i(\boldsymbol{o}_{k+1}, \boldsymbol{c_{k+1}}))$
14:        Update the critic by minimizing the loss:

$$L = \frac{1}{S} \sum_k ((y_k - V(\boldsymbol{s}_k))^2 + (\hat{y}_k - Q(\boldsymbol{s}, \boldsymbol{w}_k))^2)$$

15:        Update the actor along with the encoder using sampled policy gradient:

$$\nabla_{\boldsymbol{\theta}_u} J(\cdot, \boldsymbol{\theta}_u) = \mathbb{E}_{\boldsymbol{s} \sim \rho^\pi, \boldsymbol{u} \sim \pi}[\nabla_{\boldsymbol{\theta}_u} \log \pi(\boldsymbol{u}|\boldsymbol{o}, \boldsymbol{c})[r + \gamma V_{\theta_c}(\boldsymbol{s}') - V_{\theta_c}(\boldsymbol{s})]]$$

16:        Update scheduler using sampled policy gradient:

$$\nabla_{\boldsymbol{\theta}_{\mathrm{wg}}} J(\boldsymbol{\theta}_{\mathrm{wg}}, \cdot) = \mathbb{E}_{\boldsymbol{w} \sim \mu}[\nabla_{\boldsymbol{\theta}_{\mathrm{wg}}} \mu(\boldsymbol{o}) \nabla_{\boldsymbol{w}} Q_{\theta_c}(\boldsymbol{s}, \boldsymbol{w})|_{\boldsymbol{w} = \mu(\boldsymbol{o})}]$$

17:        Update target network parameters:

$$\theta'_{\mathrm{wg}} \leftarrow \tau \theta_{\mathrm{wg}} + (1 - \tau)\theta'_{\mathrm{wg}}$$

$$\theta'_c \leftarrow \tau \theta_c + (1 - \tau)\theta'_c$$

18:     **end for**
19: **end for**

---

## B   DETAILS OF ENVIRONMENTS AND IMPLEMENTATION

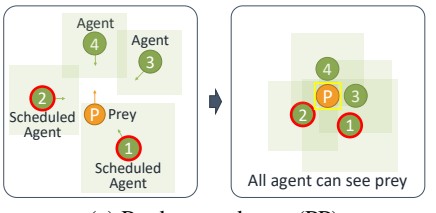
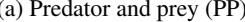

(a) Predator and prey (PP)

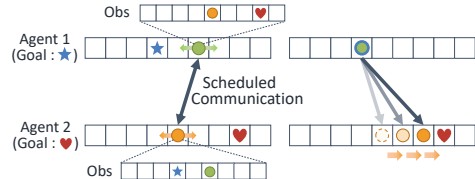

(b) Cooperative communication and navigation (CCN)

Figure 7: Illustrations of the experimental environment

### B.1 ENVIRONMENTS: PP AND CCN

**Predator and prey** We assess SchedNet in this predator-prey setting as in Stone & Veloso (2000), illustrated in Figure 7a. This setting involves a discretized grid world and multiple cooperating predators who must capture a randomly moving prey. Agents' observations include position of themselves and the relative positions of the prey, if observed. The observation horizon of each predator is limited, thereby emphasizing the need for communication. The termination criterion for the task is that all agents observe the prey, as in the right of Figure 7a. The predators are rewarded when the task is terminated. We note that agents may be endowed with different observation horizons, making them heterogeneous. We employ four agents in our experiment, where only agent 1 has a $5 \times 5$ view while agents 2, 3, and 4 have a smaller, $3 \times 3$ view. The performance metric is the number of time steps taken to capture the prey.

**Cooperative communication and navigation** We adopt and modify the cooperative communication and navigation task in Lowe et al. (2017), where we test SchedNet in a simple one-dimensional grid as in Figure 7b. In CCN, each of the two agents resides in its one-dimensional grid world. Each agent's goal is to arrive at a pre-specified destination (denoted by the square with a star or a heart for Agents 1 and 2, respectively), and they collect a joint reward when both agents reach their target destination. Each agent has a zero observation horizon around itself, but it can observe the situation of the other agent. We introduce heterogeneity into the scenario, where the agent-destination distance at the beginning of the task differs across agents. In our example, Agent 2 is initially located at a farther place from its destination, as illustrated in Figure 7b. The metric used to gauge the performance of SchedNet is the number of time steps taken to complete the CCN task.

### B.2 EXPERIMENT DETAILS

Table 1 shows the values of the hyperparameters for the CCN and the PP task. We use Adam optimizer to update network parameters and soft target update to update target network. The structure of the networks is the same across tasks. For the critic, we used three hidden layers, and the critic between the scheduler and the action selector shares the first two layers. For the actor, we use one hidden layer; for the encoder and the weight generator, three hidden layers each. Networks use rectified linear units for all hidden layers. Because the complexity of the two tasks differ, we sized the hidden layers differently. The actor network and the critic network for the CCN have hidden layers with 8 units and 16 units, respectively. The actor network and the critic network for the PP have hidden layers with 32 units and 64 units, respectively.

Table 1: List of hyperparameters

| Hyperparameter | Value | Description |
|---|---|---|
| training step | 750000 | Maximum time steps until the end of training |
| episode length | 1000 | Maximum time steps per episode |
| discount factor | 0.9 | Importance of future rewards |
| learning rate for actor | 0.00001 | Actor network learning rate used by Adam optimizer |
| learning rate for critic | 0.0001 | Critic network learning rate used by Adam optimizer |
| target update rate | 0.05 | Target network update rate to track learned network |
| entropy regularization weight | 0.01 | Weight of regularization to encourage exploration |

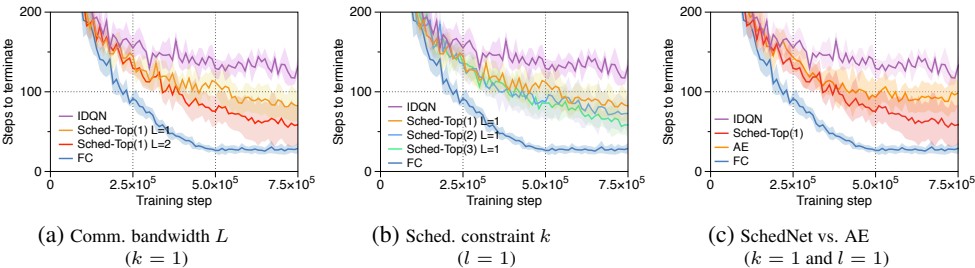

(a) Comm. bandwidth $L$
$(k = 1)$

(b) Sched. constraint $k$
$(l = 1)$

(c) SchedNet vs. AE
$(k = 1$ and $l = 1)$

Figure 8: Performance evaluation of SchedNet. The graphs show the average time taken to complete the task, where shorter time is better for the agents.

## C  ADDITIONAL EXPERIMENT RESULTS

### C.1  PREDATOR AND PREY

**Impact of bandwidth ($L$) and number of schedulable agents ($K$)**   Due to communication constraints, only $k$ agents can communicate and scheduled agents can broadcast their message, each of which has a limited size $l$ due to bandwidth constraints. We see the impact of $l$ and $k$ on the performance in Figure 8a. As $L$ increases, more information can be encoded into the message, which can be used by other agents to take action. Since the encoder and the actor are trained to maximize the shared goal of all agents, they can achieve higher performance with increasing $l$. In Figure 8b, we compare the cases where $k = 1, 2, 3$, and FC in which all agents can access the medium, with $l = 1$. As we can expect, the general tendency is that the performance grows as $k$ increases.

Table 2: Performance with/without encoder

| FC | SchedNet -Top(1) | Schedule w/ auto-encoder |
|---|---|---|
| 1 | 2.030 | 3.408 |

**Impact of joint scheduling and encoding**   To study the effect of jointly coupling scheduling and encoding, we devise a comparison against a pre-trained auto-encoder (Bourlard & Kamp, 1988; Hinton & Zemel, 1994). An auto-encoder was trained ahead of time, and the encoder part of this auto-encoder was placed in the Actor's ENC module in Figure 1. The encoder part is not trained further while training the other parts of network. Henceforth, we name this modified Actor "AE". Figure 8c shows the learning curve of AE and other baselines. Table 2 highlights the impact of joint scheduling and encoding. The numbers shown are the performance metric normalized to the FC case in the PP environment. While SchedNet-Top(1) took only 2.030 times as long as FC to finish the PP task, the AE-equipped actor took 3.408 times as long as FC. This lets us ascertain that utilizing a pre-trained auto-encoder deprives the agent of the benefit of joint the scheduler and encoder neural network in SchedNet.

**What messages agents broadcast**   In Section 4.1, we attempted to understand what the predator agents communicate when performing PP task where $k = 1$ and $l = 2$. In this section, we look into the message in detail. Figure 9 shows the projections of the messages generated by the scheduled agent based on its own observation. In the PP task, the most important information is the location of the prey, and this can be estimated from the observation of other agents. Thus, we are interested in the location information of the prey and other agents. We classify the message into four classes based on which quadrant the prey and the predator are included, and mark each class with different colors. Figure 9a shows the messages for different relative location of prey for agents' observation, and Figure 9b shows the messages for different locations of the agent who sends the message. We can observe that there is some general trend in the message according to the class. We thus conclude that if the agents observe the prey, they encode into the message the relevant information that is helpful to estimate the location of the prey. The agents who receive this message interpret the message to select action.

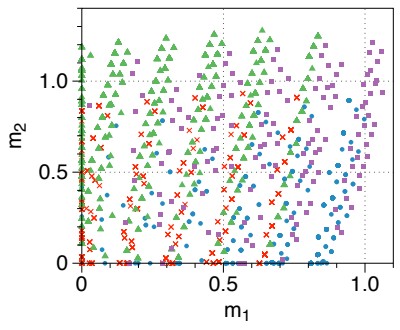 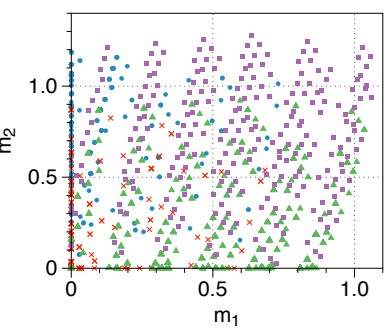

(a) Messages for different relative location of pery.

(b) Messages for different location of agent.

Figure 9: Projection of encoded messages into 2D plane in PP.

## C.2 PARTIAL OBSERVABILITY ISSUE IN SCHEDNET

In MARL, partial observability issue is one of the major problems, and there are two typical ways to tackle this issue. First, using RNN structure to indirectly remember the history can alleviate the partial observability issues. Another way is to use the observations of other agents through communication among them. In this paper, we focused more on the latter because the goal of this paper is to show the importance of learning to schedule in a practical communication environment in which the shared medium contention is inevitable.

Enlarging the observation through communication is somewhat orthogonal to considering temporal correlation. Thus, we can easily merge SchedNet with RNN which can be appropriate to some partially observable environments. We add one GRU layer into each of individual encoder, action selector, and weight generator of each agent, where each GRU cell has 64 hidden nodes.

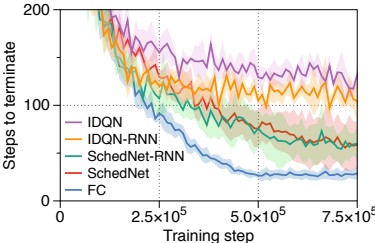

Figure 10 shows the result of applying RNN. We implement IDQN with RNN, and the results show that the average steps to complete tasks of IDQN with RNN is slightly smaller than that of IDQN with feed-forward network. In this case, RNN helps to improve the performance by tack-

Figure 10: Impact of applying RNN ($k = 1$ and $l = 2$)

ling the partial observable issue. On the other hand, SchedNet-RNN and SchedNet achieve similar performance. We think that the communication in SchedNet somewhat resolves the partial observable issues, so the impact of considering temporal correlation with RNN is relatively small. Although applying RNN to SchedNet is not really that helpful in this simple environment, we expect that in a more complex environment, using the recurrent connection is more helpful.

## C.3 COOPERATIVE COMMUNICATION AND NAVIGATION

**Result in CCN** Figure 11 illustrates the learning curve of 200,000 steps in CCN. In FC, since all agents can broadcast their message during execution, they achieve the best performance. IDQN and COMA in which no communication is allowed, take a longer time to complete the task compared to other baselines. The performances of both are similar because no cooperation can be achieved without the exchange of observations in this environment. As expected, SchedNet and DIAL outperform IDQN and COMA. Although DIAL works well when there is no contention constraint, under the contention constraint, the average number of steps to complete the task in DIAL(1) is

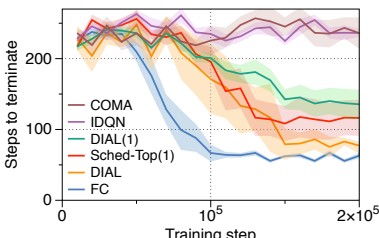

Figure 11: Comparison with other baselines

larger than that of SchedNet-Top(1). This result shows the same tendency with the result in PP environment.

## D    SCHEDULER FOR DISTRIBUTED EXECUTION

**Issues.** The role of the scheduler is to consider the constraint due to accessing a shared medium, so that only $k < n$ agents may broadcast their encoded messages. $k$ is determined by the wireless communication environment. For example, under a single wireless channel environment where each agent is located in other agents' interference range, $k = 1$. Although the number of agents that can be simultaneously scheduled is somewhat more complex, we abstract it with a single number $k$ because the goal of this paper lies in studying the importance of considering scheduling constraints.

There are two key challenges in designing the scheduler: *(i)* how to schedule agents in a distributed manner for decentralized execution, and *(ii)* how to strike a good balance between simplicity in implementation and training, and the integrity of reflecting the current practice of MAC (Medium Access Control) protocols.

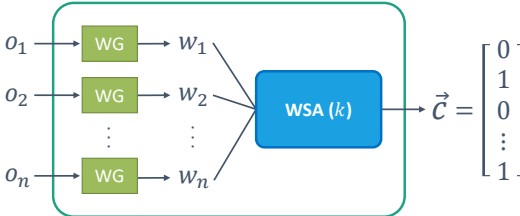

Figure 12: Proposed scheduling architecture. Each agent $i$ calculates its scheduling weight $w_i$ from weight generator (WG), and the corresponding scheduling profile $c \in \{0,1\}^n$ is determined by the scheduling algorithm (k) (WSA(k)), satisfying the condition $||c||_1 = k$.

**Weight-based scheduling**    To tackle the challenges addressed in the previous paragraph, we propose a scheduler, called weight-based scheduler (WSA), that works based on each agent's individual weight coming from its observation. As shown in Figure 12, the role of WSA is to map from $w = [w_i]_n$ to $c$. This scheduling is extremely simple, but more importantly, highly amenable to the philosophy of distributed execution. The remaining checkpoint is whether this principle is capable of efficiently approximating practical wireless scheduling protocols. To this end, we consider the following two weight-based scheduling algorithms among many different protocols that could be devised:

- *Top(k).* Selecting top $k$ agents in terms of their weight values.
- *Softmax(k).* Computing softmax values $\sigma(w)_i = \frac{e^{w_i}}{\sum_{j=1}^n e^{w_j}}$ for each agent $i$, and then randomly selecting $k$ agents with probability in proportion to their softmax values.

Top($k$) can be a nice abstraction of the MaxWeight (Tassiulas & Ephremides, 1992) scheduling principle or its distributed approximation (Yi et al., 2008), in which case it is known that different choices of weight values result in achieving different performance metrics, *e.g.*, using the amount of messages queued for being transmitted as weight. Softmax($k$) can be a simplified model of CSMA (Carrier Sense Multiple Access), which forms a basis of 802.11 Wi-Fi. Due to space limitation, we refer the reader to Jiang & Walrand (2010) for detail. We now present how *Top(k)* and *Softmax(k)* work.

### D.1    CARRIER SENSE MULTIPLE ACCESS (CSMA)

CSMA is the one of typical distributed MAC scheduling in wireless communication system. To show the feasibility of scheduling *Top(k)* and *Softmax(k)* in a distributed manner, we will explain the variant of CSMA. In this section, we first present the concept of CSMA.

**How does CSMA work?**    The key idea of CSMA is "listen before transmit". Under a CSMA algorithm, prior to trying to transmit a packet, senders first check whether the medium is busy or idle, and then transmit the packet only when the medium is sensed as idle, *i.e.*, no one is using the

channel. To control the aggressiveness of such medium access, each sender maintains a backoff timer, which is set to a certain value based on a pre-defined rule. The timer runs only when the medium is idle, and stops otherwise. With the backoff timer, links try to avoid collisions by the following procedure:

- Each sender does not start transmission immediately when the medium is sensed idle, but keeps silent until its backoff timer expires.
- After a sender grabs the channel, it holds the channel for some duration, called the holding time.

Depending on how to choose the backoff and holding times, there can be many variants of CSMA that work for various purposes such as fairness and throughput. Two examples of these, *Top(k)* and *Softmax(k)*, are introduced in the following sections.

### D.2    A VERSION OF *Distributed Top(k)*

In this subsection, we introduce a simple distributed scheduling algorithm, called *Distributed Top(k)*, which can work with SchedNet-Top($k$). It is based on CSMA where each sender determines backoff and holding times as follows. In SchedNet, each agent generates the scheduling weight $w$ based on its own observation. The agent sets its backoff time as $1 - w$ where $w$ is its schedule weight, and it waits for backoff time before it tries to broadcast its message. Once it successfully broadcasts the message, it immediately releases the channel. Thus, the agent with the highest $w$ can grab the channel in a decentralized manner without any message passing. By repeating this for $k$ times, we can realize decentralized Top($k$) scheduling.

To show the feasibility of distributed scheduling, we implemented the Distributed Top($k$) on Contiki network simulator (Dunkels et al., 2004) and run the trained agents for the PP task. In our experiment, Top($k$) agents are successfully scheduled 98% of the time, and the 2% failures are due to probabilistic collisions in which one of the colliding agents is randomly scheduled by the default collision avoidance mechanism implemented in Contiki. In this case, agents achieve 98.9% performance compared to the case where Top($k$) agents are ideally scheduled.

### D.3    OCSMA ALGORITHM AND *Softmax(k)*

In this section, we explain the relation between *Softmax(k)* and the existing CSMA-based wireless MAC protocols, called oCSMA. When we use *Softmax(k)* in the case of $k = 1$, the scheduling algorithm directly relates to the channel selection probability of oCSMA algorithms. First, we explain how it works and show that the resulting channel access probability has a same form with *Softmax(k)*.

**How does oCSMA work?**    It is also based on the basic CSMA algorithm. Once each agent generates its scheduling weight $w_i$, it sets $b_i$ and $h_i$ to satisfy $w_i = \log(b_i h_i)$. It sets its backoff and holding times following exponential distributions with means $1/b_i$ and $h_i$, respectively. Based on these backoff and holding times, each agent runs the oCSMA algorithm. In this case, if all agents are in the communication range, the probability that agent $i$ is scheduled over time is as follows:

$$s_i(\boldsymbol{w}) = \frac{\exp(w_i)}{\sum_{j=1}^{n} \exp(w_j)}.$$

We refer the readers to Jang et al. (2014) for detail.

