# OpenReview forum: "Learning to Schedule Communication in Multi-agent Reinforcement Learning"
_ICLR.cc/2019/Conference_

### Official Review · AnonReviewer2 · 2018-11-03
**Generally ok, but hard to gauge significance of this work**

**Rating:** 7
**Confidence:** 2

**Review:**

# overview
This paper focuses on multi-agent reinforcement learning tasks that require communication between the agents, and further presupposes that the communication protocol is bandwidth constrained and contentious so that a scheduling mechanism is necessary.  To address this they introduce a new learned weighting scheme based scheduler and distributed actor, centralized critic based architecture which is evaluated on a couple of communication driven multi-agent tasks.

The two evaluation tasks had their bandwidth artificially constrained, and SchedNet time to convergence was shown to fall somewhere between having no communication and full communication, and somewhat better than a purely round-robin based scheduling scheme, which doesn't seem particularly informative.  From this it is difficult to assess the significance of the contributions.

# pros
* communication in multi-agent scenarios is an important aspect to consider, and this work shines a spotlight on scenarios in which bandwidth is constrained.
* general presentation fairly clear and easy to read

# cons
* Would have been more impactful to focus experiments on real-world scenarios in which bandwidth is constrained and naturally contentious

# other comments
* pg. 2 related work, suspect you meant to call out Foerster et al 2017b in second reference not Foerster et al 2017a twice.

---

> ### Author Response · Authors · 2018-11-16
> **Contribution of our work**
>
> Apologies for the delay in response. We hope that our answers clarified our contribution and thank the reviewer again.
>
> 1. Contribution of our work
>
> We would like to highlight that our work’s contribution is to provide a need of considering communication scheduling issues in MARL under shared medium contention constraint. To this end, first, we take a shared medium contention issue into consideration in MARL problem which could be a critical problem when deploying the algorithm in the real world, where the communication among the agents are typically done in wireless. Due to the contention, only a limited number of agents are able to simultaneously use the medium. Thus, scheduling is the major concern in this setting. Secondly, we propose the framework to learn how to schedule the communication to achieve the common goal of the application. We employ a weight based scheduling which can cover a various scheduling set, and train a neural network to learn how to set the weight to maximize the common shared reward in MARL. We think that our proposed neural network may not be the best one, which may require to include other smart modules for better performance (e.g., RNN structure and explicit consideration of message senders). However, raising this scheduling issue to the MARL community seems valuable, which, we hope, motivates further future work.
>
> Comparing SchedNet with a round robin scheme is meaningful, since we can see the power of jointly considering the effect of a given task and scheduling, which is actually why we propose SchedNet. The objective of conventional scheduling algorithms in wireless research community is often simple, e.g., fairness or throughput, which may not be closely related to the common reward depending on the application. Round robin based scheduling scheme is the typical example of scheduling algorithms for fairness. The experiment results show that if the agents learn how to schedule themselves by considering their common goal for better cooperation, then they can get a higher common reward.
>
> To the best of our knowledge, SchedNet is the first work that considers the shared medium issues in MARL, and we expect that there will be other further works considering communication scheduling.
>
>
> 2. Implications of our simulation environment
>
> We think Predator and Prey (PP) environment can be an illustrative example of real-world multi-agent cooperation scenarios. One example is the cooperative task of UAVs (Unmanned Aerial Vehicles) where UAV matches to predator in PP environment. In general, UAVs are equipped with wireless communication, and their communication environment is as follows.
>
> UAVs are controlled by sequential command whose interval is very short (e.g., 0.02 sec). If there are a lot of UAVs, then it is difficult to allow all UAVs to have the chance to communicate, because only one agent can broadcast the message at a time. Given such short decision period, only a limited number of agents can communicate before making a decision of every command.
>
> Another constraint is about bandwidth. If we use the unlicensed band radio (e.g., WiFi, ZigBee), the bandwidth is limited due to the regulation on the transmission power. This means that the bandwidth constraint has to be taken into account as well as contention constraint when designing the communication protocol. DIAL explicitly considered the bandwidth constraint into their design, but to the best of our knowledge, there is no prior work which tackles the contention constraint.
>
> Although our experiments were done with challenging MARL tasks, yet all evaluations were conducted in a simulated environment. We plan to extend our work into real-world scenario such as UAV and real-time robot control applications, which can be more impactful and challenging.

---

> > ### Comment · AnonReviewer2 · 2018-12-08
> > **upward revision based on response**
> >
> > Thank you for taking the time to write a detailed response further clarifying your contributions and outlining a couple of appropriate real-world evaluation directions for future work.  Taking this additional information into account I've adjust my review score from 6 to 7.

---

### Official Review · AnonReviewer3 · 2018-11-03
**Well written, easy to follow.**

**Rating:** 8
**Confidence:** 5

**Review:**

The authors present a setting of MARL communication where only a number of agents can broadcast messages in a shared and limited bandwidth channel. The paper is well written and easy to follow, and the authors run an extensive number of baselines to illustrate the contributions.

Comments:

1) It's not clear to me how do the authors tackle partial observability without the use of recurrent connections or time-steps?

2) Do the agents know if they were chosen to be broadcasted at the previous timestep?

3) Many times it's important to know who sent the message, do the agents share this information?

---

> ### Author Response · Authors · 2018-11-16
> **Partial observability, schedule awareness, and sender identification**
>
> Apologies for the delay in response. Thank you for your interest in our article
>
> Q1. It's not clear to me how do the authors tackle partial observability without the use of recurrent connections or time-steps?
>
> In MARL, there are two ways to tackle partial observability issues. First, using RNN structure to indirectly remember the history can alleviate the partial observability issues. Another way is to use the observations of other agents through communication among them.
>
> The goal of this paper is to show the importance of learning to schedule in a practical communication environment in which the shared medium contention is inevitable. Thus, we focused more on the latter in which efficient communication including scheduling helps to enlarge the observation of a single agent well and mitigates the partial observability issue. We comment that each agent uses its own scheduling history as the input of scheduling weight generator in our experiment. Although we did not use a typical RNN structure, we think that using history information of scheduling helps to address partial observability issues.
>
> Enlarging the observation through communication is somewhat orthogonal to considering temporal correlation, so merging both approaches is possible.
> SchedNet does not assume any specific neural network structure as an intrinsic/critical part of its algorithm, where feed-forward NNs is just one choice.
>
> We have applied RNN to both IDQN — in which no communication is allowed — and SchedNet, and run more experiments in PP environment. We have updated the result in appendix C.2 of our modified manuscript. The results show that the average steps to complete tasks of IDQN with RNN is slightly smaller than that of IDQN with feed-forward network. In this case, RNN helps to improve the performance by tackling the partial observable issue. On the other hand, SchedNet-RNN and SchedNet achieve similar performance. We think that the communication in SchedNet somewhat resolves the partial observable issues. Although applying RNN to SchedNet is not really that helpful in this simple environment, we expect that in a more complex environment, using the recurrent connection is more helpful.
>
>
> Q2. Do the agents know if they were chosen to be broadcasted at the previous timestep?
>
> Yes, the agents know whether they are scheduled in the previous timestep in our experiment. This is reasonable because the previous scheduling information of itself is easy to access in a real environment based on some practical a wireless scheduling protocol.  For example in 802.11 WiFi protocol, each agent has a decrementing counter initialized randomly, and check whether the channel is clean (clear channel assessment) when the counter turns zero. From this information, each agent can immediately know if the transmission has been done or it is waiting until the channel is clear. As we mentioned above, this information of scheduling histories is usefully used to determine the scheduling weight, which in turn decides whose observation is important or not.
>
>
> Q3. Many times it's important to know who sent the message, do the agents share this information?
>
> We agree. In our experiments, we assume that the agents do not know explicitly who sent the message. However, agents are trained to generate different messages which might include the unique feature of each agent implicitly. Thus, if identifying the sender is important, the agent learns to encode the sender’s identification information into a message and to decode the message to extract this information.
>
> Also, although the agent does not have the explicit knowledge of who sent the message, the scheduler of SchedNet informs whose message is more important via learning weights. Thus, this helps the agents to consider the message from the agents who have the most important observation.
>
> There can be some cases where the agents’ identifying the sender might be helpful. SchedNet can also be easily extended to this case. The information about sender can be used as the input to the action selector (AS), which can be helpful for better performance. If it is not useful, then AS will be trained not to use this information.

---

> > ### Comment · AnonReviewer3 · 2018-11-19
> > **Response**
> >
> > Q1. Thank you for the clarification and the update. The results may depend on the complexity of the setting, but in my opinion, anything without short / long term memory is that it's a weak architecture choice for tackling partial observability.
> >
> > Q2. Thanks.
> >
> > Q3. Although I find it quite far away, but since the 802.11 was referenced many times, I think that even if messages are broadcasted the identity is a common characteristic for most communication protocols. That coueld also help the performance of the proposed model.

---

> > > ### Author Response · Authors · 2018-11-20
> > > **Thanks for the comments**
> > >
> > > We agree with the opinion of reviewer’s about partial observability. Although we got the experiment result where RNN does not significantly improve the performance of SchedNet in PP environment, applying RNN to SchedNet can be pivotal in tackling partial observability issue in more complicated environments. We plan to examine SchedNet in more complex and realistic environments as a future topic.
> > >
> > > As mentioned by the reviewer, in many protocols it is not difficult to get the identification information of sender. Even when agent does not use this information, we have seen the impact of scheduling communication. We expect that the usage of the information about sender is helpful for better performance, and that it is worth doing more experiments to gauge its helpfulness.

---

### Official Review · AnonReviewer1 · 2018-11-03

**Rating:** 7
**Confidence:** 3

**Review:**

The authors present a study on scheduling multi-agent communication. Specifically, the authors look into cases where agents share the same reward and they are in a partially observable environment, each of them with different observations. The main contribution of this work is that authors provide a model for communication scheduling for dealing with cases where only a certain number of agents is allowed to communicate.

The paper is very clear, positions the work very well in the literature of MARL and communication. The authors perform experiments in two environments and include a number of reasonable baselines (e.g., adapted DIAL for top(k))  as well as the full-communication upper bound.
The authors moreover provide a nice analysis on the messages in the predator-pray experiment.

My only concern is that authors report "DIAL(1) performs worse than SchedNet-Top(1)".  However, Figure 3a clearly shows that Dial(1) to be within the variance of Sched-Top(1) -- from this it's not clear that the null hypothesis can be rejected. The authors should probably verify this with a statistical test cause at the moment their claim is unsupported. Moreover, why Figure 3c does not contain the same models as Figure 3a (e.g., DIAL appears to be missing)?

---

> ### Author Response · Authors · 2018-11-16
> **Statistical testing and baseline comparison**
>
> Apologies for the delay in response. We would like to thank the reviewers for evaluating our manuscript.
>
> 1. Statistical testing
>
> As the reviewer commented, the experiment results in our submitted version of the paper does not clearly support our claim which is “SchedNet-Top(1) outperforms DIAL(1)” due to the large variance. We have toned down the claim as “the average number of steps to capture the prey in DIAL(1) is larger than that of SchedNet-Top(1)” in Section 4.1.
>
> For the reviewer’s convenience, we report a p-value of 0.075 with just 25 samples collected from a comparative simulation analysis of DIAL(1) and Sched-Top(1). The p-value shows a decreasing tendency with even just 25 samples, thereby tending towards a non-negligible performance gap that is not attributable to random factors alone.
>
> We comment that the performance comparison with DIAL(1) is not for showing that SchedNet is better than DIAL(1), but mainly for highlighting that scheduling should be considered to get a higher reward. We comment that SchedNet is not intended for competing with DIAL but a complementary one. We believe that adding our idea of agent scheduling makes prior works more practical and valuable.
>
>
> 2. Baseline comparison
>
> We have updated the results that show the comparison with other baselines in appendix C.3 of our modified manuscript. The result shows a similar tendency with the result in PP depicted in Figure 3a. IDQN and COMA in which no communication is allowed, take a longer time to complete the task compared to other baselines with communication. DIAL works better than SchedNet-Top(1), however, under the contention constraint the average number of steps to complete the task in DIAL(1) is larger than that of SchedNet-Top(1).

---

### Meta-Review · Area_Chair1 · 2018-12-15

**Confidence:** 5
**Recommendation:** Accept (Poster)

**Metareview:**

The authors present a learnt scheduling mechanism for managing communications in bandwidth-constrained, contentious multi-agent RL domains. This is well-positioned in the rapidly advancing field of MARL and the contribution of the paper is both novel, interesting, and effective. The agents learn how to schedule themselves, how to encode messages, and how to select actions. The approach is evaluated against several other methods and achieves a good performance increase. The reviewers had concerns regarding the difficulty of evaluating the overall performance and also about how it would fare in more real-world scenarios, but all agree that this paper should be accepted.